

# Assessment of the Hype Model for Simulation of Water and Nutrients in the Upper uMngeni River Catchment in South Africa

Jean N. Namugize[1], Graham P.W. Jewitt[1,2], David Clark[1] and Johan Strömqvist[3]

[1] Centre for Water Resources Research, School of Agriculture, Earth and Environmental Sciences, University of KwaZulu-Natal, Scottsville, 3209, South Africa

[2] Umgeni Water Chair of Water Resources Management, School of Engineering, University of KwaZulu-Natal, Scottsville, 3209, South Africa

[3] Swedish Meteorological and Hydrological Institute (SMHI), SE-60176, Norrköping, Sweden

*Correspondence to*: Jean N. Namugize (najoannes@yahoo.fr)

**Abstract.** Most studies considering water quality pollution in the upper reaches of the uMngeni Catchment have relied on the physical grab sampling of water and the subsequent laboratory analysis of chemical determinants. However, this provides limited spatial and temporal information. Thus, the objectives of this study are to assess the capability of the Hydrological Predictions for the Environment (HYPE) model in simulating streamflow, dissolved inorganic nitrogen (DIN) and total phosphorus (TP), in the fast developing uMngeni Catchment in KwaZulu-Natal province, South Africa. The model was set up and calibrated, following a stepwise approach and then validated. Results indicated that the simulation of discharge is most sensitive to the parameters related to evapotranspiration and the water-holding capacity of the soil, while DIN and TP are affected by plant uptake and initial pools of nutrients. DIN is also affected by denitrification. Runoff was captured well during the calibration (1989-1995) and validation periods (1961-1999), with a Nash-Sutcliffe efficiency (NSE) greater than 0.0 in eight of the nine stations and a Pearson's correlation coefficient (r) of > 0.5 at all the sub-catchments. High streamflow events were represented well, low streamflows were over-simulated. The accumulative streamflows were over-predicted in the downstream sub-catchments, with an absolute percentage of bias (PBIAS) of > 25 %. The transport and dynamics of DIN and TP vary differently and they are driven by hydrological and biochemical processes. The concentration of TP follows the pattern of the streamflow, whereas DIN shows an inconsistent variation. The values of DIN decrease from upstream to downstream, while the TP values increase from the headwaters to the outlet of the catchment. Agricultural activities were found to be the largest source of DIN, while the TP is mainly ascribed to the point sources of pollution.

**Key words:** denitrification; HYPE model; nutrients; uMngeni Catchment; water quality model



# 1 Introduction

A key aspect of the global water quality challenge arises from the eutrophication of fresh, marine and coastal waters, and it is ascribed to the world's rising population, a change in the demographics and land use/land cover patterns, water consumption, urbanisation and climate change effects (Falkenmark, 1990; UNEP-GEMS/Water, 2008; Zimmerman et al., 2008). Globally, the agricultural sector was reported to use approximately 70 % of the freshwater in 2000 and this rate was over 85 % in the developing countries of Asia and Africa (UN-Habitat, 2003; FAO, 2004). The deterioration of water quality is a worldwide

concern and is highly accentuated in developing countries, due to a growing demand for water and limited funding-sources for water research, treatment and restoration (Prepas and Charette, 2004).

Runoff from agriculture, as well as human and industrial waste, are known to be the largest sources of the increased concentrations of nitrogen and phosphorus in surface and groundwater (UNEP, 2010). To better understand the factors

affecting water quality decline in waterbodies, the primary nutrients responsible for eutrophication, investigations on the interactions between the hydrological cycle, soil types, climatic conditions and land use activities are of the utmost importance (Tong and Chen, 2002). This cannot be achieved by following the routine water quality monitoring programmes, or relying on grab and/or continuous river sampling. Therefore, process-based hydrological and water quality models with varying levels of complexity, have been used to provide useful information on catchment responses in terms of nutrient loading, owing to land

use activities, soil type, crop management and climatic conditions (Thirel et al., 2015).

Depending on the size and location of the basin, the accessibility to the sampling sites, the cost and time of physical collection and the analysis of samples, water quality models can be integrated into monitoring plans, or be used alone (Loucks and Van Beek, 2005). However, high uncertainties in model simulations are recognised (Loucks and Van Beek, 2005; Moriasi et al.,

2007; Dayyani et al., 2010). A number of hydrological models, such as the ACRU-NPS agro-hydrological modelling system (Campbell et al., 2001; Smithers and Schulze, 2004), the Soil and Water Assessment Tool (SWAT) and its extensions (Arnold et al., 1998; Gassman et al., 2007; Arnold et al., 2011), the Better Assessment Science Integrating point source and Non-point Source of pollution (BASINS) (Di-Luzio et al., 2002), the HBV-NP model (Andersson et al., 2005) and the Hydrological Predictions for the Environment (HYPE) (Lindström et al., 2010) have been developed.


Research studies in uMngeni Catchment in South Africa have reported on the nutrient, sediment and bacteriological pollution of the river and its dams (Hemens et al., 1977; Breen, 1983; Graham, 2004; Quayle et al., 2010; Van Ginkel, 2011; GroundTruth, 2012; Lin et al., 2012; Gakuba et al., 2015; Matongo et al., 2015; Ngubane, 2016), which were ascribed to rapid urbanisation, increases of informal settlements with poor sanitation, the expansion of agricultural lands, livestock farming,

bulk atmospheric deposition, dysfunctional sewage networks and the use of phosphorus detergents. The concentrations of total nitrogen and total phosphorus that exceed the recommended South African eutrophication threshold limits of 0.6 mg L$^{-1}$ and



0.055 mg L$^{-1}$, respectively, were noted in the rivers and dams of the upper reaches of the uMngeni Catchment (DWAF, 2002). As a result, the occurrence of cyanobacterial algae and increased levels of Chlorophyll-a have been reported in the Midmar and Albert Falls Dams (Matthews, 2014; Matthews and Bernard, 2015).


A locally-developed rainfall-runoff model, ACRU has previously been applied in the uMngeni Catchment, to simulate the flow of water, sediment, *Escherichia coli* and phosphorus (Kienzle et al., 1997) and its sub-model, ACRU-NPS in the Mkabela sub-catchment (41 km$^2$), to simulate nitrate, phosphorus and sediments (Kollongei and Lorentz, 2015). In addition, the pollution loading estimator (PLOAD) was used in the simulation of the total phosphorus export-coefficient at quaternary

catchment level (Dabrowski et al., 2013). The focus here, is to apply a readily available "off the shelf" model and to test its applicability in this setting. Thus, following the bilateral collaboration in water resources management between the Department of Water and Sanitation of South Africa and the Ministry of Environment and Energy of Sweden, the successful application of the HYPE model in simulating the transport and dynamics of water, nitrogen and phosphorus in catchments of different scales (Strömqvist et al., 2012; Jiang et al., 2014; Jomaa et al., 2016); and the lack of in-stream processes in ACRU-NPS. This

study sought to test the capabilities of HYPE model. Focus is on simulation of nutrients in a fast-developing catchment, where the human conversion of natural vegetation to other land uses is substantial and where the increasing nutrient content of water is an issue. This catchment has also very different climatic and physiographic conditions to Sweden. Therefore, the overall aim of this study was to test HYPE in the upper reaches of the uMngeni Catchment, an area which is typical of rapidly developing conditions of southern Africa. This involved: (i) the simulation of streamflow and the concentration of dissolved

inorganic nitrogen (DIN equates $NH_4 + NO_3$) and total phosphorus; and (ii) providing insight into sources of the increased concentrations of DIN and TP and their spatial distribution in the catchment.

## 2 Methodology

### 2.1 Study area

The case study catchment in the upper reaches of the uMngeni Catchment cover a surface area of 1653 km$^2$ (29.78° and 30.42°

E and 29.23° and 29.63° S), in the KwaZulu-Natal province, South Africa (Fig. 1). The mean annual precipitation in the catchment ranges between 400 and 1000 mm per annum, with most of the rains falling in summer months (October to April) and the occurrence of occasional rains in winter. The mean annual temperature in the catchment ranges between 12 °C and 14 °C, with the minimum temperature during winter months of June and July and the maximum temperature during December-February. The potential evaporation measured at the A-pan in the area, ranges between 1600 mm and 1800 mm per annum

(UW, 2016).

Based on the topography, soil type, land use, altitude, water management, inter-transfer, water sampling and the gauging sites, the uMngeni Catchment has been demarcated into 13 sub-catchments, known as the Water Management Units (WMUs) by



the Department of Water Affairs and Forestry (Warburton, 2012). The flow of the major river draining the catchment from the

Drakensberg Mountains to the Indian Ocean, the uMngeni River is regulated by four large dams, from upstream to downstream viz. the Midmar, Albert Falls, Nagle and Inanda. These dams are the sources of drinking water for over five million inhabitants of the catchment, as well as for agriculture water use (Kienzle et al., 1997; Hay, 2017). The Lions and Karkloof Rivers are the major tributaries of the uMngeni River, upstream of the Midmar and Albert Falls Dams, respectively. Over 300 farm dams are scattered through the catchment. The Msunduzi River passes through Pietermaritzburg, the major city of KwaZulu-Natal

Province, and joins the uMngeni River downstream of the Nagle Dam. The catchment has lost over 15 % of its natural vegetation over the past two decades, due to rapid urbanization and the expansion of agricultural lands and forest plantations (Jewitt 2012; Mauck and Warburton, 2013; Jewitt et al., 2015a; Namugize et al., submitted).

In 2004, water demands began to exceed the supply in the uMngeni Catchment, water transfer schemes were constructed from

Mooi River to the uMngeni River and others are planned for the future (DWAF, 2008; UW, 2013, 2016). This involved the construction of the Mearns Weir and Spring Grove Dams and the transfer pipelines which outfall into Mpofana River, which is the tributary of Lions River upstream of the Midmar Dam. It was expected that by 2016, the two transfer schemes could supply approximately 4.5 $m^3$ $s^{-1}$ on a continuous basis (UW, 2016). The water quality of the Upper uMngeni River and large dams has been monitored by Umgeni Water (UW) since its establishment in 1974. Recent data have reported a deterioration

of water quality in the upper reaches of the uMngeni catchment, as indicated by an increase in the concentrations of phosphorus, nitrogen and *Escherichia coli (E.coli)*. This was attributed to the growth of informal settlements in the Mphophomeni Township, expansion of agricultural activities, effluent discharges from wastewater treatment and runoff from urban areas of Howick and Hilton (DWAF, 2008; Taylor et al., 2016). In addition, water of the Spring Grove Dam and Mearns Weir (in Mooi River Catchment), the sources of water which is transferred to Midmar Dam contains high concentrations of nutrients and

*E.coli* counts which could affect the receiving Mpofana River (DWAF, 2008). Umgeni Water is also the supplier of bulk water to municipalities to support different users (the population, agriculture and industry). Previous analyses have highlighted the Midmar and Albert Falls Catchments, which are part of the focus of this study, as important sources of water supply of the whole uMngeni Catchment system (Jewitt et al., 2015b).

120                                                   <Figure 1>

## 2.2 Description of the HYPE model

The HYPE model is a physically-based, semi-distributed water quantity and water quality model, developed by the Swedish Meteorological and Hydrological Institute (SMHI). The first version was developed during the period 2005-2007, but the

development of the model is still ongoing and the model source code is open. The model simulates at a daily time-step the transport and turnover of water and nutrients in the soil, rivers, lakes and dams and has been primarily applied in large





catchments, with areas ranging between 0.35 million km² and 8.8 million km² (Lindström et al., 2010; Strömqvist et al., 2012; Arheimer et al., 2015; Hundecha et al., 2016). However, the model also works well at small scales (for example, a catchment of 99.5 km²) (Jiang et al., 2014; Andersson et al., 2015; Pers et al., 2016). The model's establishment was based on previous

Swedish models, HBV and HBV-NP (Lindström et al., 2010; Yin et al., 2016). HYPE was developed to assist in overcoming the problem of eutrophication, which was a major issue of water quality in Sweden (Strömqvist et al., 2012). In the model, the catchment is divided to sub-catchments, which, in turn, are divided into classes, based on a combination of soil texture and land cover types (SLC). These SLCs are referred as hydrological response units, which are the smallest units of hydrological calculations in the model (Lindström et al., 2010; Strömqvist et al., 2012).


Above the ground, the model simulates snow, evapotranspiration, glaciers, rivers, lakes and routing. Within the soil, water content is computed for each of a maximum of three layers. The major biological and chemical transformation of nitrogen and phosphorus in land, streams and lakes, namely, immobilisation, absorption/desorption, plant uptake, primary production, mineralisation, denitrification, sedimentation and resuspension, are taken into account. The model simulates nitrogen

(dissolved inorganic (DIN) and organic (ON)) and phosphorus (particulate (PP) and soluble reactive phosphorus (SRP)), and total nitrogen (TN) and total phosphorus (TP) are calculated as the sum of their respective fractions. In addition, organic carbon and conservative tracers, such as chloride and ¹⁸O, can also be modelled (Lindström et al., 2010; Jiang and Rode, 2012; Pers et al., 2016). A schematic illustration of the processes and routing of water and nutrients (N and P) in the HYPE model is shown in Fig. 2. The major model parameters and algorithms are fully described in Lindström et al. (2010). Comprehensive

updated documentation on model processes, source code and update versions are free for download at the webpage http://hypecode.smhi.se/.

The HYPE model has been applied in many catchments of the northern hemisphere in the simulation of water quantity and pollutants (nitrogen, phosphorus and organic carbon). Examples of this include the whole of Sweden (Strömqvist et al., 2012;

Arheimer et al., 2015; Pers et al., 2016), a European multi-basin study (Donnelly et al., 2016; Hundecha et al., 2016), simulation of nutrient losses in Germany (Jiang and Rode, 2012), the Baltic Sea (Donnelly et al., 2011), the assessment of climate change effects on water resources in the whole of India (Pechlivanidis et al., 2015) and in few studies on the African continent (Andersson et al., 2014; Andersson et al., 2015), but limited to West Africa.

<Figure 2>

## 2.3 Model set-up

The application of the HYPE model to the upper reaches of the uMngeni Catchment has involved the collection of input data from various sources. The mandatory data for the model included climate data (daily temperature and precipitation), land use





types, soil type information, crop type, agricultural practices, dam data, water quality data and information on the point sources
of pollution in the catchment.

The model was set up for the period 1950-1999, for which a complete dataset of temperature and precipitation is available.
The 53 sub-catchments were delineated in the study area. This mirrors the process for configuration of ACRU developed by
Warburton et al. (2010) for water quality studies. The DEM was used to calculate the elevation-means, the elevation standard
deviation, the slope-means and the slope standard deviation for each sub-catchment. Soil information has been disaggregated
from the Land Type Map of South Africa, which resulted in five classes of soil texture i.e. loam, sand clay loam, sand loam,
silt loam and no texture, as shown on Fig. 1d (Land Type Survey Staff, 1972-2006). The reclassification of the National Land
Cover (NLC) 2000 dataset resulted in twelve land use classes that can affect water quality *viz.* indigenous forest,
thicket/bushland, natural grassland, planted grassland, planted forest, bare rock/soil, cultivated dryland, cultivated irrigated
land, residential, waterbody, mines/quarries and wetlands. Information on crop types was deduced from the NLC 2000 (Fig.
1c).

The combination of soil types, land use and crop type information in ArcGIS 10.1 led to 68 SLCs, which have been reduced
to 53 SLCs after aggregating the SLCs that have similar properties (Yin et al., 2016). Each SLC is not coupled to any
geographical location, but it represents a fraction of the sub-catchment area (Lindström et al., 2010; Jiang et al., 2014). The
model sums the simulated flow of water and nutrients from each SLC within a sub-catchment and routes that and water from
upstream sub-catchments to the sub-catchment outlet. The flow path of water and nutrients follows the flow direction between
the sub-catchments and ends up at the catchment outlet. A sketch illustrating the direction of flow is presented in Fig. 3.

Two soil layers extracted from the Land Type map of South Africa, their corresponding depths and the drain depth were
defined (Land Type Survey Staff, 1972-2006). However, a third thick soil layer was added during the calibration of the model.
In HYPE model, evapotranspiration is assumed to occur from the top two upper soil layers, decreases with soil depth and is at
potential rate if the water content of soil exceeds field capacity (fc) (Lindström et al., 2010). For this study, the potential
evapotranspiration (PET) was calculated using the modified Jensen-Haise/McGuinness models (Jensen and Haise, 1963) Eq.
(1). This PET model is less data demanding (needs only air temperature and solar radiation) and provides the better simulations
of streamflows in comparison to the other PET models (Jensen and Haise, 1963; Oudin et al., 2005).

$$\text{PE} = \frac{R_e}{\lambda \rho}\left(\frac{T_a + K_2}{K_1}\right) \; if \; T_a + \; K_2 > 0 \tag{1}$$

PE=0                     otherwise



Where: PE is the rate of potential evapotranspiration (mm day$^{-1}$), $R_e$ = extra-terrestrial radiation (M J m$^{-2}$ day$^{-1}$) which depends on latitude and Julian day, $T_a$ = mean daily air temperature (°C) is a function of the Julian day for a given location, λ = latent

heat flux (assumed equal to 2.45 M J kg$^{-1}$) and ρ represents the density of water (kg m$^{-3}$). The constants $K_1$ (°C) and $K_2$ (°C) are the model fixed parameters which are fixed for each rainfall/runoff model.

<Figure 3>

In estimating effluent generation from the households not connected to municipal wastewater plants for each sub-catchment, the ArcGIS tools have been used to overlay sub-catchments and administrative boundaries. An equal distribution of the population within a sub-catchment was assumed, and the fraction of the population per sub-catchment was calculated, using the 1996 population census data in four local municipalities covering the upper reaches of uMngeni Catchment namely, Impendle, uMngeni, Mooi-Mpofana and uMshwati (SSA, 2016). Domestic wastewater generation was estimated to be 150

litres per capita per day, which is the average domestic wastewater discharge established by the Department of Public Works (DPW) (DPW, 2012).

In the model set-up, rating curves of the two lakes (the Midmar and Albert Falls Dams) were defined in the LakeData file, while the average daily water abstraction and water transfer were added as negative and positive point sources, respectively,

in the model file PointSourceData. Information on the fraction of water withdrawn from groundwater for irrigation in each sub-catchment was set in the MgmtData file. Data on the amount of fertilisers and the timing was obtained from the literature and from the ACRU-NPS Model, as well as information on the soil water holding capacity and the number of soil layers and their corresponding thicknesses. Concentrations of nitrogen and phosphorus in the effluent discharge, rivers and daily streamflow data were acquired from the South African Department of Water and Sanitation (DWS) webpage and from UW.

More details on the source of the model inputs are presented in Table 1.

## 2.4 Model calibration and evaluation

The HYPE model has over one hundred parameters, most of them are either general, land use or soil-type dependent (Lindström et al., 2010). Consequently, a step-wise manual calibration was carried out, starting with the parameters affecting the

hydrological simulation and followed by the water quality calibration (Strömqvist et al., 2012). The calibration involved an iterative adjustment of the model's input parameters, until the model outputs represented the hydrological behaviour of the catchment as far as possible, through verification against observed data. As the water quality component of the model has been largely used in the climatic and physiographic conditions of the northern hemisphere, many of the water quality variables were adjusted to the conditions of the study area.




From the nine gauging stations (Sub-catchments 7, 27, 28, 1510, 2511, 33, 3411, 35 and 45), only three stations 2511 (DWS number U2H006), 1510 (DWS ID number U2H007) and 7 (DWS ID number U2H013) were selected for the calibration of both hydrology and nutrients (DIN and TP). These stations have a complete data set of daily discharges and weekly nutrient concentrations, they are located upstream of the two large dams of the study area, they are less influenced by urban
development, they cover approximately 60 % of the catchment surface and they have different land use/land cover types. The model was calibrated for the period 1989-1995 for both the streamflow and water quality. The streamflows and water quality were validated for the period 1961-1999 and 1995-1999, respectively, following the hydrological year, which starts on the 1$^{st}$ October and ends on 30$^{th}$ September. A number of parameters that affect the generation of runoff at the outlet of each sub-catchment, were manually adjusted.


The most important parameters that affected the simulation of runoff in the model are divided into two categories: those linked to land use, for example, the crop coefficient for potential evapotranspiration (Kc); those that are more general, like the factor for calculating the soil water limit for evapotranspiration (lp); and others related to the flow and retention of water in the soil *viz*. the runoff coefficient for the first and second soil layers (rrcs1 and rrcs2), the wilting point (wcwp), the effective porosity
(wcep) and the field capacity (wcfc).

For water quality, the HYPE model simulates immobile and dissolved pools of nitrogen and phosphorus, which are influenced by the soil properties and external sources of nutrients. In this study, the concentration of DIN was largely controlled by the initial values of the pools of humus (humusN) and the fast-organic nitrogen in the soil (fastN), as well as the amount of N in
manure (mn1) and fertilisers (fn1) applied to the soil. The process of transformation of nitrogen in the soil and water, such as the crop uptake function of nitrogen (up1), the denitrification in local/main rivers (denitwl/denitwrm), in lakes (denitwrl) and in soil (denitrilu), influences the outputs of DIN. The outputs of TP also depend on the initial soil pools of phosphorus (humusP, partP, fastP), crop uptake function (up1), as well as the amount of P in fertilisers (fp1), in manure (mp1) and in decaying plants (resP). The soil type dependent factors namely, the phosphorus leaching factor (freuc), soil erodibility (soilerod) and soil
resistence to erosion (soilcoh) also have slight effects on the output of P. An adjustment of the rate of sedimentation of particulate P in lakes (sedPP) and the sedimentation and resuspension of PP in the local watercourse also have minor effects.

The performance of the model was evaluated, using commonly-used statistics in hydrology, such as the Nash Sutcliffe Efficiency (NSE) (Nash and Sutcliffe, 1970), which compares the residual variance of the simulated and the measured data,
the Pearson's correlation coefficient (r), which describes the degree of collinearity between the modelled and the measured data, and the percent bias (PBIAS) (Arnold et al., 2012; Yin et al., 2016), which represents the capability of the model to capture the water balance (Moriasi et al., 2007; Yin et al., 2016). The NSE ranges between -∞ and 1, with a perfect fit when NSE = 1. An acceptable performance value of NSE between 0.0 and 1.0 was suggested for a daily time-step, whereas the NSE value < 0.0 indicates an unacceptable performance.



The optimal value of PBIAS is 0, whereas the positive and negative PBIAS values indicate the model over-estimation and under-estimation of the flow, respectively (Gupta et al., 1999; Jomaa et al., 2016). Moriasi et al. (2007) highlighted that, for a monthly time-step, the absolute PBIAS value ≤ 25% and NSE > 0.5 for streamflow can be judged as satisfactory. The Pearson's correlation coefficient ranges between -1 and +1, depending on whether a negative or a positive relationship exists between the simulated and the observed values, while an r = 0 shows non-existent of relationship between the variables. Moreover,

graphical visualisation techniques were also used on time-series data, to examine the capability of the model to represent the peak and the base flow events (Moriasi et al., 2007; Crout et al., 2008). These evaluation criteria were used in other applications of the HYPE model (Strömqvist et al., 2012; Jomaa et al., 2016; Yin et al., 2016). Moreover, another evaluation criterion, comparing the average daily simulated and observed flows, was used. An acceptable simulation is achieved when the percentage difference between the observed and simulated daily streamflows is less than 15 %. This criterion was used in

former studies in the catchment (Warburton et al., 2010). A list of all parameters used in the calibration as well as their physical meaning, is presented in Appendix A.1.

The equations of NSE, r and PBIAS are described below:

$$NSE = 1 - \frac{\sum_n (s - o)^2}{\sum_n (o - \bar{o})^2}$$

(2)

$$PBIAS = \frac{\sum_n (s - o) * 100}{\sum_n o}$$

(3)

(4)

$$r = \frac{\sum_{i=1}^{n} (s - \bar{s}).(o - \overline{o})}{\sqrt{\sum_{i=1}^{n} (s - \bar{s})^2 . \sum_{i=1}^{n} (o - \bar{o})^2}}$$

Where (s) stands for simulated, (o) for observed, (s) for simulated, ō for average observed value and (s̄) for average simulated.


## 3 Results and discussion

### 3.1 Simulation of streamflows

During the calibration period (1989-1995), the HYPE model has satisfactorily simulated water flow at seven sub-catchments out of nine (NSE ≥ 0.0). The best model performances were noted at the outlets of sub-catchments 7, 28, 2511, 35 and 3411

(NSE ~ 0.6), with unacceptable simulations at Sub-catchment 33 (NSE < 0.0). A linear relationship between simulated and observed values was noted (r ranging between 0.5 and 0.8) in all nine Sub-catchments, during both the calibration and



validation periods. In the validation period, NSE remained in the same range, with a slight improvement at Sub-catchment 45 (NSE = 0.5) and with the exception of the Sub-catchment 33 (NSE = -0.5). As suggested by Moriasi et al. (2007), the shorter time intervals, the poorer the model simulations. Thus, the simulation of streamflows at a monthly step provided acceptable

simulation results (NSE > 0.5) at six out of the nine stations (Table 2). The NSE value <0.0 for daily and monthly simulations at Sub-catchment 33, suggests that observed average streamflows are better predictors than simulated values, which indicates a generally poor performance of the model at this sub-catchment. This poor performance at Sub-catchment 33 may be attributed to possible errors in the observed streamflow and a short time-span of observed streamflows. The good simulations at outlet of sub-catchment (45) were achieved after adding manually the daily water release from Albert Falls Dam into the simulated

streamflows. This resulted in good representation of the water balance in the model (PBIAS of -4.8 % for calibration and -7.7 % for validation periods). Furthermore, these under-simulations of the flows in downstream sub-catchments (3411 and 35) during the validation period (Table 2) can also be ascribed to the simplified evapotranspiration processes in HYPE model (Strömqvist et al., 2012; Jiang et al., 2014).

The values of PBIAS indicated a good performance of the model in four sub-catchments, for the validation period, with some improvement for a monthly time-step (PBIAS ≤ 25 %). An over-simulation of streamflow was noted during the calibration period at the uMngeni at Petrus Stroom (7), at the outflow of the Midmar Dam (3411) and at the uMngeni in Howick (35), with good simulation at 45, where the absolute values of PBIAS were less than 25 % (Table 2). During the validation period, there was alternation between the over-simulation of flows in some sub-catchments (1510) and under-simulation in the others

(7, 2511, 3411and 35).

The high flow events were captured well in some instances, while in the others, a model deviation was noted (Fig. 4). A generalised over-simulation of base flow was identified in the three representative sub-catchments (Fig. 4). The model has consistently predicted high streamflows in the summer months (December to February), which is the period of when much of

the rain falls in the catchment and low streamflows in the winter months. Moreover, the non-inclusion of a large number of farm dams within the catchment could have increased the level of uncertainties in the model outputs (Venohr et al., 2005). Inconsistencies in representing the low flow events that occurred between August and November were noted, with their over-simulation in some instances (for example, at Sub-catchment7 on 5/10/1992 and 20/09/1993, at Sub-catchment 1510 on 20/09/93 and 22/10/1992, at Sub-catchment2511 on 18/09/1993 and 16/09/1995), as well as their under-simulation (for

example, at Sub-catchment 7 on 18/8/1990, at Sub-catchment1510 on 22/9/1991 and at Sub-catchment2511 on 10/09/1991). This has further implications on the simulation of nutrients. Non-representation of some peak flows in the HYPE model was reported by other studies, as a result of the low precision of the model in capturing flash flood events, which are caused by heavy and short-term rainfall events (Jiang et al., 2014).



However, the high percentage of over-prediction of streamflows (30 %) recorded at Petrus Stroom (7) during the calibration period, decreased to 12 % during the validation period (Table 3). This is consistent with findings of Donnelly et al. (2016) who reported on the poor representation of daily hydrographs and extremes in mountainous regions. Otherwise, the HYPE model has provided acceptable simulations of streamflow in eight sub-catchments, as indicated by the statistics presented in Table 2. The modelled daily simulated streamflows exceeded the measured values by less than 15 % for the validation period (1961-

1999), which confirms the acceptability of our results at the three sub-catchments used for the calibration (Table 3). Our percentages of the over- or under-estimation of streamflows are comparable to the findings of Warburton et al. (2012), which are 7.9 % at Sub-catchment 7, 9.9 % at Sub-catchment 1510 and 13.05 % at Sub-catchment 2511 for average daily simulations (for the period 1987-1998). However, the HYPE Model provided better representations of high flow events than ACRU, despite its over-prediction of low flows. For this study, an over-simulation of low flows could be ascribed to the recession

coefficients, which control the downward movement of water in the soil. Moreover, this model was developed in Swedish post-glacial hydrological conditions, characterised by small depth to the groundwater table, shallow soils, a large number of lakes and a low rate of evapotranspiration, which are completely different to the areas included in this study (Lindström et al., 2010; Strömqvist et al., 2012).

<Figure 4>


When the model was evaluated on monthly mean values, many improvements were noted in the model's performance, where the NSE value for calibration and validation at four sub-catchments (7, 2511, 28 and 3411) was 0.7, as presented in Table 2. Furthermore, the graphic visualisation showed good fits between the monthly mean values and observed streamflows, and the high and low streamflow events are represented well (Fig. 5). The severe drought in 1983 and the 1987 floods that occurred in

the catchment, were effectively captured by the model (Fig. 5).

<Figure 5>

### 3.2 Simulation of water quality

We have simulated TN, DIN, ON, PP, SRP and TP in the model. However, in the section below only the concentration of DIN and TP are presented, as no historical data on TN were available. Thus, TN was not calibrated. Moreover, in the model input data, the PP was calculated as the difference between TP and SRP and most of data points on SRP were below the detection limit of the analytical methods used at the UW laboratories (5 µgP $L^{-1}$). In the evaluation of the daily simulations of water quality outputs, we have avoided using the NSE coefficient, since the errors are compared to the variance of the concentrations

data, which were collected on a weekly sampling-frequency. The graphical visualisation techniques and the Pearson's correlation coefficient r were used in this case. This approach has also been followed in other applications of HYPE model (Strömqvist et al., 2012).





A high variability in the concentrations of DIN in the simulations was noted, as presented at Fig. 6. For example, some spikes
in concentrations were identified during the period of low flows, while the others were noted during high flows events. This
was noted at all three sub-catchments during the calibration and validation periods. The occurrence of large concentrations of
DIN was noted during the winter months and in the first months of summer. These high values of DIN in winter could be
ascribed to the possible release of nitrates from ammonium-nitrate fertilisers in the agricultural lands, with a substantial
leaching of $NO_3$ into the water. Moreover, the contribution of bush burning, a common practice in the area during the winter
months, as well as the first rains falling at the beginning of summer, cannot be ignored, as they transport the accumulated
nutrients into the landscape.

<Figure 6>

An under-simulation of DIN concentrations was found at the three sub-catchments in February 1991 and December 1995,
when flood events occurred in the catchment, but runoff was captured well. The model seems to result in dilution of DIN
during floods while observed data seems to show that that the floods are mobilising the DIN. In general, in cases where high
concentrations of DIN were measured in period of high peak flows, the model under-estimated these consistently across the
three sub-catchments (Fig. 6). Despite these mismatches between the simulated and observed DIN, an overall under-simulation
of DIN was noted during the validation and calibration periods.

The daily concentration of TP followed the pattern of the simulated streamflows. This indicated that much of runoff in the
catchment coincides with the high contents of TP in water during the period from December-March (Fig. 7). Therefore, some
peaks of the observed values greatly exceeded the simulated values, and vice versa. The possible reasons could be that: (1)
water quality samples emanate from a grab sampling programme, which cannot capture all high flow events, (2) samples are
usually collected in the morning, while much of the rain in the catchment falls in the afternoon (Ngubane, 2016), (3) the model
itself provides average daily concentrations of nutrients, indicating that short high flow events are not captured well, which
may result in the release of high levels of P and N from farm dams, (4) phosphorus release by the soil and in-stream erosion,
since the soil-phosphorus cohesion properties in the catchment limit its release; and (5) a number of data points of the measured
TP were below the detection limits ($< 15 \mu gP L^{-1}$) of the analytical methods used at Umgeni Water's laboratories (for example,
from May to October 1994 for Petrus Stroom and from April to September 1991 for the Karkloof River).

In comparison to DIN, the TP outputs were satisfactory, as the percentage of under-/over-simulation at Sub-catchments 7,
1510 and 2511 were -4 %, -9 % and +6 %, respectively, for the period 1989-1999. However, under-simulation of DIN at Sub-
catchments 7, 1510 and 2511 were 24 %, 39 % and 18 %, respectively, which indicates the poor performance of the model for
DIN.



<Figure 7>

## 3.3 Evaluation statistics of water quality

The performance of water quality simulations indicated that for TP, ten of the twelve sub-catchments have positive Pearson's correlation coefficients. The highest r of ~ 0.4 was noted at Sub-catchment 33 (representing the Mthinzima outflow to the Midmar Dam). However, the strength of this linear relationship declines during the validation period, with a maximum value of 0.28 at Sub-catchment 7. Simulated DIN correlated well with the measured values, compared to TP, especially during the calibration period (r = 0.55 at sub-catchment 1210). A generally poor performance of the model in the sub-catchments

downstream of the Midmar Dam was identified (Table 4).

## 3.4 Seasonal distribution of flows and nutrients

An assessment of the seasonal distribution of runoff and the concentration of DIN and TP was undertaken for the period between 1989 and 1999. The results showed some mismatches between the simulated and the observed DIN. They also

indicated that the routing and biogeochemical transformation of DIN in streams is largely driven by the hydrological processes in the HYPE model. The lowest simulated concentrations of DIN were noted in January-February, as mentioned in Fig. 6, which is a period of high runoff generation in the catchment (Fig. 8a and 8c). These discrepancies may be ascribed to the high nitrogen uptake by crops, an increase in the denitrification process, due to the high temperature of the water, as well as the possible dilution of the water. Another reason that may explain this is the high turbidity that characterises the water in the

catchment, which can speed up the denitrification process. In this regard, these findings are similar to those reported by Jiang et al. (2014) and Jomaa et al. (2016) in nested mesoscale catchments in central Germany.

In contrast to the DIN, the concentrations of TP followed the streamflow patterns concurrently (Fig. 7, 8b and 8c). At these three sub-catchments (7, 1510 and 2511), the model under-predicted the peak concentration of TP (January-February), while

during the low flow months (May to September), an over-prediction of TP was noted at 2511, with fluctuations in TP at Sub-catchments 7 and 1510 (Fig. 8b). In general, there has been a good fit between the simulated and observed TP and streamflows in the catchment, which indicates the capability of the model to represent the runoff and TP generation in the catchment. The seasonal distribution of runoff was reproduced well, with the highest flow in January-February and the lowest in September, as also reported by Kienzle et al. (1997). Moreover, in-stream erosion and soil erosion which are not modelled in HYPE, are

reported to contribute significantly to the phosphorus concentration (Pers et al., 2016). In addition, it appeared that HYPE lagged the observations of nutrients (especially TP) and streamflow at sub-catchments 7, 1510 and 2511 (Fig. 8b and 8c). This could be attributed to incremental contribution of upstream sub-catchments (river lengths) and high retention rate of TP in the catchment (Breen, 1983; Lindström et al., 2010).



<Figure 8>


## 3.5 Distribution of DIN and TP in the catchment

The map of the average-weighted annual distribution of DIN concentrations relates the large values of DIN in sub-catchments with intensive agricultural activities, such as upstream of Sub-catchment 1611 (Fig. 9a). It also shows the sub-catchments with significant point sources of pollution (33 and 37) and it highlights a substantial retention of DIN in the Midmar Dam. The

highest concentrations of DIN were identified in sub-catchments with a high density of informal settlements (29, 30, 32, 33 and 37), as well as to the waste water treatment works (Fig. 9a). Thereafter, the sub-catchments with a dominance of agricultural lands i.e. commercial planted forests (43), as well as irrigated and dryland cultivation (upstream of sub-catchment 17, 27 and 28) (Fig. 1c and 9a). In contrast, very high concentrations of TP are identified in the sub-catchments located in the stretch between the Midmar and Albert Falls Dams (Fig. 9b). This indicates that TP originates mainly from the point sources

of pollution, while DIN comes from both the diffuse and point sources.

These findings confirm the high phosphorus-adsorption nature of the soil in the catchment, which reduces its leaching during runoff (Breen, 1983). Based on these results, an increase in the concentration of TP in the water of the Midmar and Alberts Falls Dams (3411 and 45) could be ascribed to outflows from the surrounding sub-catchments, supplemented by the point

sources of pollution upstream of the Albert Falls Dam, such as Cedara, St Anne's, Howick, Mpophomeni Township and many feedlots (Hudson et al., 1993; Kienzle et al., 1997; DWAF, 2008; Taylor et al., 2016).

<Figure 9>

In general, large concentrations and loads of DIN are identified in the upstream sub-catchments, where the expansion of agricultural activities, with the subsequent application of fertilisers is dominant (Ngubane, 2016). It is in these sub-catchments where the proportion of agricultural lands ranges between 25 % (Sub-catchment 7) and 47 % (Sub-catchment 2511). Planted forests are dominant in this area, irrigation practices are concentrated upstream of Sub-catchment 1510 (Cullis et al., 2005) and much of the runoff is generated in these three catchments. In addition, increased trends in concentration of nutrients were

reported in Spring Grove Dam and Mearns Weir, the intakes of the inter-basin transfer schemes to the sub-catchment 1210 (DWAF, 2008). Moreover, the steep slopes in the high-land areas of sub-catchments 2, 4, 6, 13 and 1210, can accelerate the transport capacity of DIN, owing to interflow, which is the major process driving the movement of nitrogen in the soil (Jiang et al., 2014). High concentrations of DIN in agricultural sub-catchments were also reported by Ahearn et al. (2005), Jiang et al. (2014), Arheimer et al. (2015) and Yin et al. (2016). Therefore, the load and concentration of DIN decreases from the



headwaters of the catchment to the outlet, due to retention in dams, in-stream retention and the possibility of a high rate of denitrification in the catchment (Fig. 10a).

A contrasting observation was noted for the load and high concentrations of TP, which are released from the major point-sources of pollution (Fig. 1 and 10b). The commonly-known hot-spots of pollution in the catchment i.e. the Mpophomeni
Township (sub-catchments 29 to 33), as well as the area between the Albert Falls and Midmar Dams, are where human activities are concentrated (4.6 to 23 kg ha$^{-1}$ year$^{-1}$ of TP). The retention of phosphorus in Midmar Dam does not have a significant effect on TP export in downstream sub-catchments. The presence of the major point sources of pollution mentioned in above paragraph, provides a full justification for this phenomenon.

<Figure 10>

The loadings of TP noted upstream of the Albert Falls Dam are consistent with the findings of another modelling study carried out in the catchment for the period 1989-1993, which reported the average annual phosphorus loading values as ranging between 0.5 and 850 kg km$^{-2}$, although they are slightly less than the findings of this study (Kienzle et al., 1997). Moreover, a
number of flood events that occurred after 1993, provided more information on such increased yields of TP. Previous research studies in the catchment (Kienzle et al., 1997; Dabrowski et al., 2013), as well as the current study, lead to a general conclusion that the area between the Midmar and Albert Falls Dams is the largest source of TP yield in the upper reaches of uMngeni Catchment.

### 3.6 The way forward on applications of the HYPE model in the upper reaches of the uMngeni Catchment

The application of the HYPE model in the upper reaches of uMngeni Catchment has involved the collection of data from various sources and no field work was carried out. Possible errors in the observed data were anticipated, which may lead to uncertainties in the model outputs. The model has represented high steamflow events very well, but low streamflows were over-simulated in most of the cases. Therefore, there is a need to improve this performance of the model, especially at Sub-catchment 7, where an over-simulation of streamflow was noted.


Overall simulations of streamflow in the area provided acceptable results, as indicated by the statistical performance criteria, where NSE > 0.0 at eight of the nine stations for a daily time-step and NSE ≥ 0.5 at six sub-catchments for a monthly evaluation. The poor performance of the model was noted in some sub-catchments downstream of Midmar Dam, which could be influenced by some urban development and the simplification of processes that drive the spatial variations of
evapotranspiration in HYPE model, the simplification of water abstraction, as well as releases from dams and inter-basin transfers in the model.



More than 100 parameters reflect the routing and dynamics of water and nutrients, and many of these are specific to the northern hemisphere climatic, topographic and hydrological conditions (for example, shallow groundwater, thick soil, many lakes, seasonality, agricultural management and water management practices). An adjustment was made to some of them, to reflect the local conditions. Of course, a number of these parameters were used by default in the model calibration and this may increase uncertainties in the model outputs. Therefore, due to the poor performance of the model, especially for nutrient simulations, a multi-objective calibration is suggested for nutrient modelling, which may overcome the problems inherent in the step-wise calibration approach followed.

**4 Conclusion and recommendations**

The HYPE model was tested in the upper reaches of the uMngeni Catchment to simulate streamflow, DIN and TP at a daily time-step, using historical climate data. The model was calibrated, following a stepwise approach. The most important factors affecting the predictions of runoff in the model were crop coefficient (Kc) (land use dependent), the recession coefficients of the two upper soil layers (rrcs1 and rrcs 2) and the variables related with the water storage of the soil (field capacity, wilting point and effective porosity). The most sensitive parameters in the simulation of DIN and TP were denitrification (denitrlu and denitwl for DIN), the initial pools of nutrients (resn, resp, humusPO, humusNO, partPO), crop uptake (up1) and the mineralisation of decay of fastN and fastP.

Results indicated that the model represented the water balance well, especially in the headwaters of the catchment where urban development activities are limited. High flow events were captured well, with a general over-simulation of base flow events. An under-estimation of streamflow was identified in the outlet sub-catchments, due to a simplified spatial variation of evapotranspiration processes in the model. However, the model has provided acceptable simulations of streamflows, and the good fits between modelled and measured values, especially at the monthly time-step, where NSE values of ~ 0.7 were noted in four out of the nine sub-catchments.

The simulations of DIN and TP in the model are largely influenced by the hydrological and biogeochemical process representations within the model, which indicate that poor performance in runoff simulations in turn affect the dynamics and transport of nutrients. Positive correlation coefficients between simulated and measured DIN and TP were noted in eight of the twelve monitoring sub-catchments. Mismatches between the simulated and observed DIN were identified during high flow events, for both the calibration and validation periods, with an overall under-simulation of DIN. The results indicate that the processes driving the loss and retention of nitrogen (denitrification and plant uptake) are intensified during the summer months (where low concentrations of DIN are noted), which is also a period of high temperatures and streamflow in the catchment. The spatial distribution of loads and the concentration of DIN showed high values in the upstream sub-catchments, where agricultural and forest lands are dominant and they decrease from upstream to downstream.




In contrast to DIN, the concentrations of TP followed the streamflow patterns, with spiked concentrations during high flow events. Good fits between simulated and observed TP were noted, which show the large contribution of soil and in-stream erosion to the transport and dynamics of TP in the area. Across the catchment, TP concentrations and loads are released from sub-catchments that have the major point-sources of pollution, and it was clearly that they increase from upstream to

downstream. The model has provided better predictions of TP, in comparison to those of DIN.

Overall, the testing of the HYPE model in simulating streamflow, DIN and TP has been successful in the upper uMngeni Catchment. The model has represented the streamflow and its seasonal variation in the area well. In addition, the model outputs of average concentrations of DIN and TP and their spatial distribution reflects the reality in the catchment. This has indicated that DIN is attributed to agricultural activities and the point sources of pollution; while TP originates from the point sources

of pollution. However, an application of HYPE in the catchment has some caveats related to:

- Simplification of the processes driving evapotranspiration in the model is a key challenge which affects the simulations of runoff in the catchment.

- Moreover, due to the simplification of inter-catchment transfer, water abstraction and release in the model, it would make sense to expand the application of the HYPE model to the greater uMngeni Catchment. This will provide a

bigger picture of water quality deterioration.

- A lack of updated climate data has limited our study on the simulation of streamflow and nutrients to a period up to 2000. Since then, many transformations have occurred in the catchment, such as the raising of the Midmar Dam wall in 2004 (Hay 2017), a shut down of the Mpophomeni waste stabilisation ponds, the construction of inter-basin water-transfer schemes, a decline of water quality in the upper reaches of the catchment (GroundTruth 2012; Ngubane 2016)

and the conversion of the natural vegetation to residential, agricultural and forest lands (Mauck and Warburton 2013; Jewitt et al., 2015a; Namugize et al., submitted). Therefore, there is a need to update the model simulations to the more current situation, so that recent developments can be reflected in the simulation. There is also an opportunity to consider scenario analysis of nutrient concentrations, as a result of land use change and land management practices.

Finally, as this study was a first step in the application of the HYPE model in the catchment, there is a need to collect additional

data that are required for nutrients simulation in future research, such as wet and dry deposition of DIN and TP, survey on fertiliser application and crop distribution, initial pools of phosphorus and nitrogen of the soil.

**Data availability**

HYPE is an open-source code model developed at Swedish Meteorological and Hydrological Institute (SMHI), Sweden, free to download at http://www. hypecode.smhi.se/. The forcing data utilised in this study were extracted from a compilation of

climate data of the uMngeni Catchment by Warburton (2010) and can be accessed by contacting the first author. All data on the chemistry of rivers and other point sources of pollution were provided by Umgeni Water (www.umgeni.co.za) and the




South African Department of Water and Sanitation (DWS). Data on streamflow and dams are freely downloadable from the DWS website (https://www.dwa.gov.za/Hydrology). Information on soil type was extracted from Schulze (2007) and the Agricultural Research Council -Soil, Climate and Water of South Africa (http://www.arc.agric.za/arc-iscw/Pages/ARC-ISCW-
Homepage.aspx). In addition, our simulation data are available on request.

**Authors contributions**

**JN Namugize** selected the water quality model, did the collection and preparation of inputs data, analysed the model outputs and wrote the manuscript.

**GPW Jewitt** provided supervision, guidance, technical advices and editorial services.

**D Clark** helped in GIS and in data collection

**J Strömqvist** provided technical expertise during initial set-up of the model, he guided the first author in model calibration and fixed all model trouble shootings.

All the co-authors commented on the manuscript and it has been edited by Dr Sharon Rees

**Competing interests**

The second author, GPW Jewitt is a member of the editorial board of the journal.

**Acknowledgments**

Funding for this research was provided by the South Africa Water Research Commission (WRC K5/2354), Umgeni Water and the Centre for Water Resources Research of the University of KwaZulu-Natal under the uMngeni Ecological Infrastructure
Partnership Project (UEIP). The Swedish Meteorological and Hydrological Institute (SMHI) in Sweden funded the first and third authors to attend the HYPE short-course training in Sweden. Alena Bartosova, Jafet C.M. Andersson and René Capell are acknowledged for providing technical assistance during the initial set-up of the model at SMHI. Special thanks to Dr Michele Warburton, for permitting us to use her compiled catchment daily rainfall and temperature datasets, as well as UW and DWS for making available information on point sources, water quality and river discharges. The South African Weather
Services (SAWS), UW and DWS are acknowledged for providing the climate data.

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





**Appendix A.1: A list of the selected parameters for model calibration and their physical meanings**

| Parameter | Unit | Physical meanings |
|---|---|---|
| *Streamflow* | | |
| Kc | - | Crop coefficient for PET model (land use dependent) |
| rrcs1 and rrc2 | $d^{-1}$ | Soil runoff coefficient for the uppermost soil layer and for the lowest soil layer |
| wcwp | - | Wilting point as a fraction (soil dependent) |
| wcfc | - | Fraction of soil available for evapotranspiration but not for runoff (soil dependent) |
| wcep | - | Effective porosity as a fraction (soil dependent) |
| *Water quality* | | |
| Denitrlu | $d^{-1}$ | Parameter for denitrification in soil layers (land use dependent) |
| Denitwl | $kg.m^{-2}.d^{-1}$ | Denitrification in local water course (general) |
| Denitwrl | $kg.m^{-2}.d^{-1}$ | Denitrification in lake (general) |
| Denitwrm | $kg.m^{-2}.d^{-1}$ | Denitrification in main watercourse (general) |
| sedon | $m\ d^{-1}$ | Sedimentation of ON in lakes (general) |
| fastNO | $mg\ m^{-3}$ | Initial concentrations of fast N in soil pool (land use dependent) |
| fastPO | $mg\ m^{-3}$ | Initial concentration of partP in soil pool (land use dependent) |
| sedpp | $m\ d^{-1}$ | Sedimentation of PP in lakes |
| degradhn | $d^{-1}$ | decay of humus to fastN (land use dependent) |
| dissolfn | $d^{-1}$ | decay of fastN to dissolved organic N (land use dependent) |
| dissolhn | $d^{-1}$ | decay of humusN to dissolved organic N (land use dependent) |
| humusNO | $mg\ m^{-3}$ | Initial concentration of humusN in soild pool (land use dependent) |
| degradhp | $d^{-1}$ | decay of humus to fastP (land use dependent) |
| dissolfp | $d^{-1}$ | decay of fastP to dissolved PP (land use dependent) |
| humusPO | $mg\ m^{-3}$ | starting concentration of humusP soil pool (land use dependent) |
| soilcoh | $kPa$ | characteristic of soil for calculation of soil erosion (soil dependent) |
| soilerod | $g\ J^{-1}$ | Characteristic of soil for calculation of soil erosion(erodibility) |
| freuc | $kg^{-1}$ | parameter in Freundlich equation (coefficient) (soil dependent) |
| Up1 | $g/(m^2\ yr^{-1})$ | Parameter for the crop's potential uptake function (logistic growth) |




**Tables**


**Table 1: Sources and types of input data for the HYPE Model in the upper reaches of the uMngeni Catchment, where CWRR = Centre for Water Resources Research; DWS = Department of Water and Sanitation; and UW = Umgeni Water**

| | Data | Data type | Source |
|---|---|---|---|
| 1 | Climatological data | Daily precipitation | (Warburton 2012) |
| | | Daily air temperature | (Warburton 2012) |
| 2 | Geographic data | Sub-basin area | (Warburton 2012) |
| | | Land use types | (NLC 2000) |
| | | Elevation/slope means | (Weepener et al., 2011b) |
| | | Hydrographical network, stream drainage depth, main river length | (Weepener et al., 2011a) |
| 3 | Dam information | Depth, regulation rules, rating curve | DWS |
| 4 | Soil data | Soil layer depth and number of horizons, soil layer thickness, soil water holding capacity | (Land Type Survey Staff 1972-2006; Schulze, 2007) |
| | | soil nutrient content (initial nutrient storage) | Literature |
| | | Soil texture | (Land Type Survey Staff 1972-2006; Schulze, 2007) |
| 5 | Water quality | Measured daily streamflow | DWS/UW |
| | | weekly/monthly nutrient concentrations (dissolved inorganic nitrogen (DIN), soluble reactive phosphorus (SRP) and total phosphorus (TP)) | Umgeni Water |
| 6 | Agricultural practices | Manure and inorganic fertilizer application, crop husbandry, timing and amount of fertilization, sowing and harvesting for the area | Literature |
| 7 | Water management | Sub-catchment fraction of irrigation Water withdrawn from the groundwater | (WARMS 2014) |
| 8 | Other source of nutrients | Flow from rural household not connected to the municipal wastewater works | Literature |
| | | Discharge and concentration of DIN, SRP and TP | UW, DWS |
| | | Atmospheric deposition | Literature |





**Table 2 : Model performance statistics for streamflows during the calibration and validation periods, where (1) and (2)**
**stand for daily and monthly time-steps, respectively**

| Sub-catchment | | Calibration | | | Validation (1) | | | Validation (2) | | |
|---|---|---|---|---|---|---|---|---|---|---|
| | ID | NSE | r | PBIAS | NSE | r | PBIAS | NSE | r | PBIAS |
| Petrus Stroom | 7 | 0.6 | 0.8 | 56.2 | 0.5 | 0.7 | -29.8 | 0.7 | 0.9 | -29.5 |
| Lions River | 1510 | 0.4 | 0.7 | 10.5 | 0.3 | 0.6 | 28.0 | 0.3 | 0.7 | 28.3 |
| Karkloof Shafton | 2511 | 0.6 | 0.8 | -3.7 | 0.5 | 0.7 | -29.8 | 0.7 | 0.9 | -31.2 |
| Gqishi | 27 | 0.5 | 0.8 | 2.2 | 0.5 | 0.8 | 2.1 | 0.6 | 0.9 | 1.3 |
| Nguklu | 28 | 0.6 | 0.8 | -5.9 | 0.6 | 0.8 | -4.4 | 0.7 | 0.9 | -4.1 |
| Mthinzima | 33 | -0.5 | 0.5 | 8.6 | -0.5 | 0.6 | 8.1 | -1.6 | 0.8 | 9.1 |
| Outflow Midmar | 3411 | 0.6 | 0.8 | 65.8 | 0.5 | 0.7 | -36.7 | 0.7 | 0.9 | -36.4 |
| uMngeni/Howick | 35 | 0.6 | 0.8 | 85.0 | 0.4 | 0.6 | -42.3 | 0.5 | 0.8 | -40.8 |
| Ouftflow Albert Falls | 45 | 0.3 | 0.7 | -4.8 | 0.5 | 0.8 | -7.7 | 0.5 | 0.8 | -12.2 |





**Table 3: Summary of daily streamflow simulations during the calibration (1989-1995) and validation (1961-1999) periods at three Sub-catchments, 7, 1510 and 2511, where c stands for the calibration period, v represents the validation period, sim for simulated and obs for observed**

| | Catchment ID (km2) | | 7 (294.01) | | 1510 (356.21) | | 2511(334.29) | |
|---|---|---|---|---|---|---|---|---|
| | | Unit | Obs | Sim | Obs | Sim | Obs | Sim |
| | Accumulative streamflow | mm | 1217.6 | 1579.3 | 788.2 | 843.8 | 1035.2 | 1054.5 |
| | Average daily streamflow | mm/day | 0.56 | 0.72 | 0.36 | 0.39 | 0.47 | 0.48 |
| c | Standard deviation | mm/day | 1.03 | 1.13 | 0.67 | 0.65 | 0.73 | 0.74 |
| | Pearson's coefficient | | 0.81 | | 0.67 | | 0.77 | |
| | % of under/over-simulation | % | -29.71 | | -7.06 | | -1.87 | |
| | Accumulative streamflow | mm | 10180.1 | 8965.8 | 6357.7 | 7301.7 | 9390.2 | 8855.7 |
| | Average daily streamflow | mm/day | 0.73 | 0.65 | 0.46 | 0.53 | 0.68 | 0.64 |
| v | Standard deviation | mm/day | 1.42 | 1.06 | 1.09 | 0.84 | 1.35 | 1.04 |
| | Pearson's coefficient | | 0.70 | | 0.59 | | 0.62 | |
| | % of under/over-simulation | % | 11.93 | | -14.85 | | 5.69 | |





**Table 4: Nutrient simulation performance indicated by a correlation coefficient (r) during the calibration and validation periods. ID represents the sub-catchment number; TP represents the total phosphorus and DIN represents dissolved inorganic nitrogen. * represents the sub-catchments with a high correlation coefficient**

| Sub-catchment | Sub-basin | Calibration (89-95) | | Validation (95-99) | |
|---|---|---|---|---|---|
| Name | ID | TP | DIN | TP | DIN |
| Petrus Stroom | 7 | 0.171 | 0.048 | 0.287* | 0.012 |
| Mpofana | 1210 | -0.285 | 0.554* | 0.127 | -0.040 |
| Lions River | 1510 | 0.208 | 0.086 | 0.266* | 0.125 |
| Inflow Midmar | 17 | 0.080 | -0.005 | -0.103 | 0.032 |
| Karkloof (upper) | 2510 | 0.012 | 0.143 | -0.004 | 0.189 |
| Karkloof at Shafton | 2511 | 0.236 | 0.235 | -0.002 | 0.035 |
| Mthinzima | 33 | 0.382* | 0.329* | 0.144 | 0.357* |
| Outflow of Midmar | 3411 | -0.037 | 0.258 | -0.038 | -0.059 |
| uMngeni/Howick | 35 | 0.101 | -0.206 | 0.227 | -0.068 |
| uMngeni d/s Howick | 3910 | 0.152 | -0.083 | 0.003 | 0.003 |
| uMngeni/Morton Drift | 40 | 0.064 | -0.320 | 0.120 | -0.155 |
| Outflow of Albert Falls | 45 | 0.197 | 0.114 | 0.064 | 0.052 |





## Figures




**Figure 1: The location of the study area in the uMngeni Catchment, KwaZulu-Natal province, South Africa (a), a digital elevation model (DEM) with 53 sub-catchments, water monitoring sites, stream flow gauges and rainfall stations (b), a land use and land cover ma for 2000 (c) and the soil type (d)**





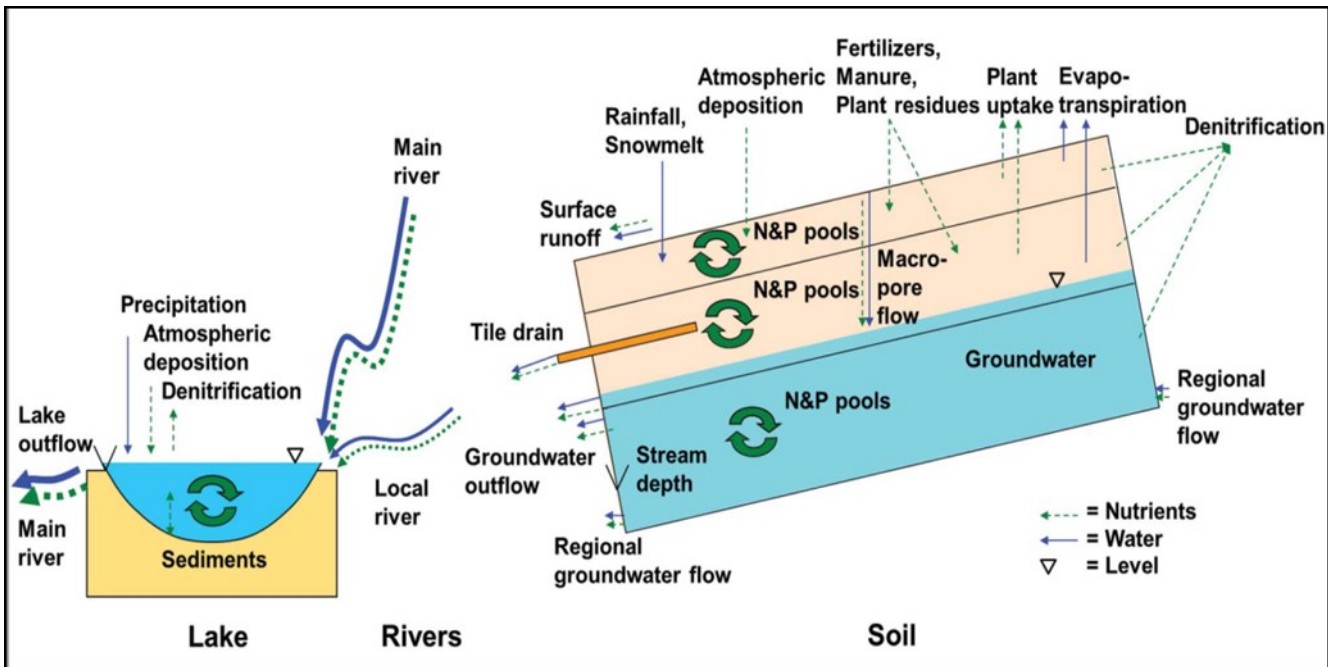

**Figure 2: Schematic illustration of nutrient transport and turnover of nutrients within a sub-basin in the HYPE model (Strömqvist et al., 2012)**





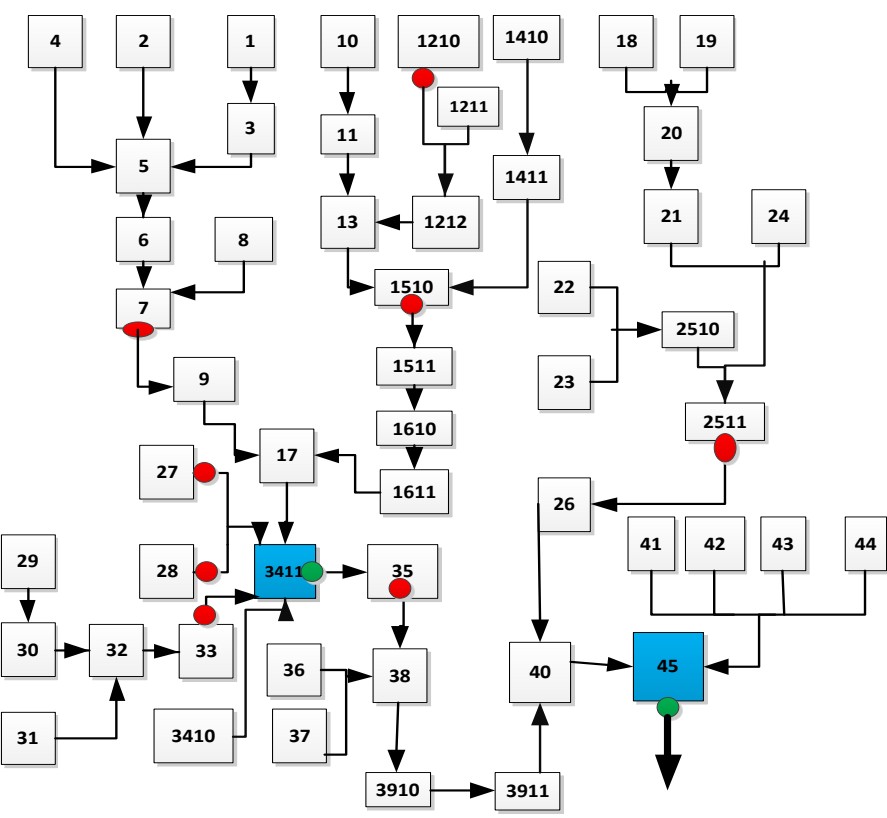

**Figure 3: Direction of flow in the catchment, where a white square represents a sub-catchment, a blue square represents a reservoir sub-catchment, a red circle represents a gauging station and a green circle for an outflow from a dam**





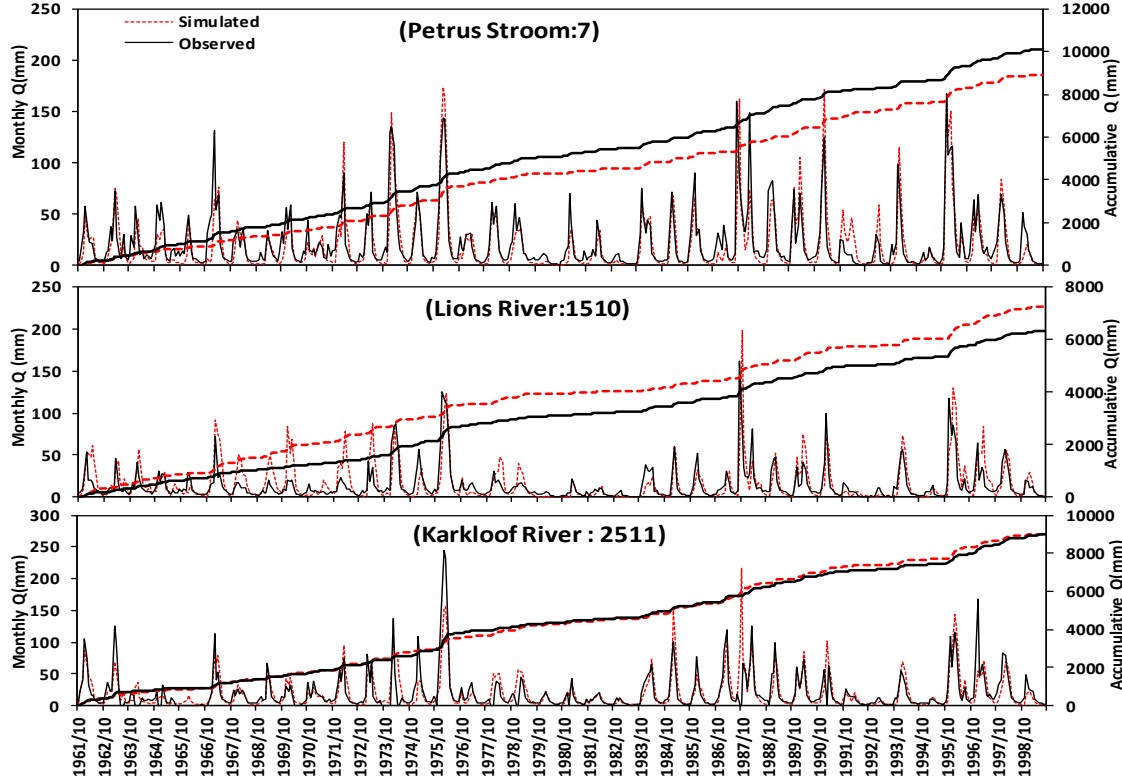

**Figure 4: Comparison of monthly totals of daily simulated and observed streamflow during the validation period**





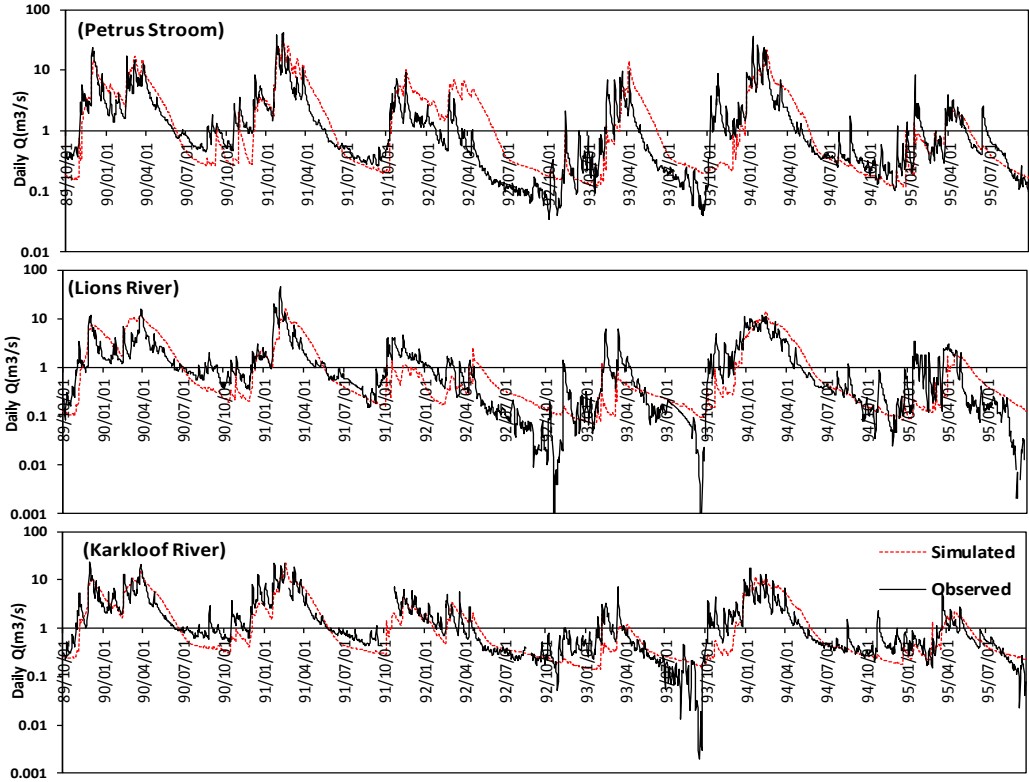

**Figure 5: Comparison of daily simulated and observed streamflows at three sub-catchments (on log scale): Petrus Stroom (7), Lions River (1510) and Karkloof River (2511), during the calibration period (1989-1995)**





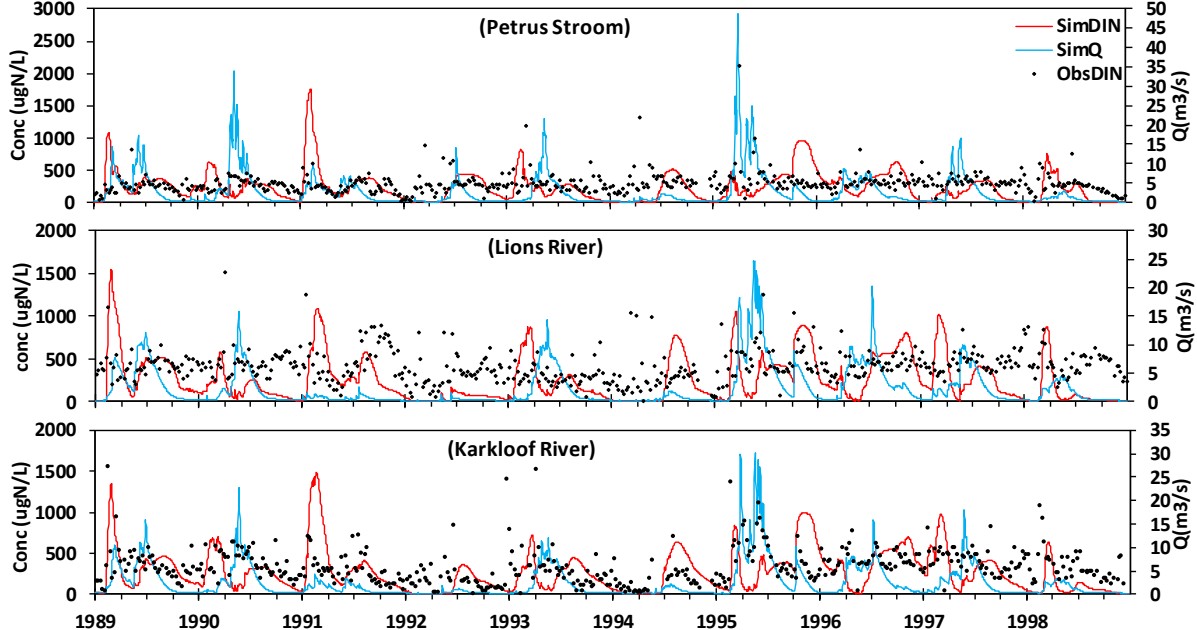

**Figure 6: Comparison of model predictions of DIN with observed data, at Petrus Stroom (7), at Lions River (1510) and at Karkloof River (2511), for the calibration (1989-1995) and validation (1995-1999) periods. The red line represents the daily simulated concentrations (µg L$^{-1}$), the blue line represents the simulated daily stream flow and the dots represent the observed weekly concentrations (µg L$^{-1}$).**





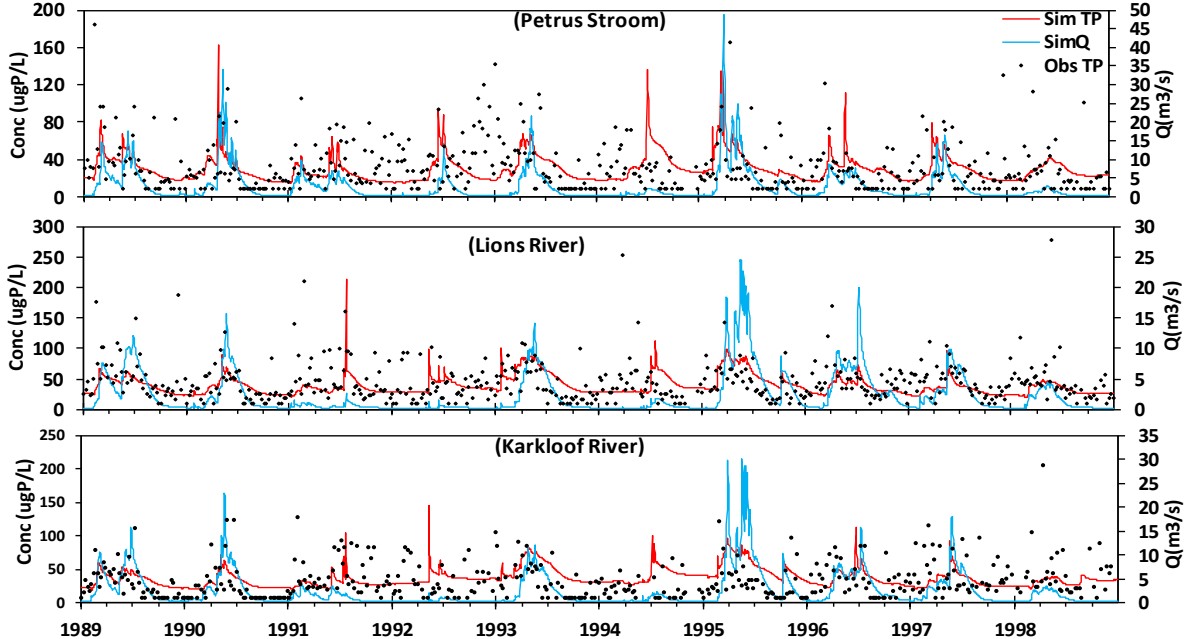

**Figure 7: Comparison of model predictions of TP with observed data, at Petrus Stroom (7), Lions River (1510) and Karkloof River (2511), for the calibration (1989-1995) and validation (1995-1999) periods. The red line represents the**

**daily simulated concentrations (μg L$^{-1}$), the blue line represents the simulated daily stream flows and the dots represent the observed weekly concentrations (μg L$^{-1}$).**





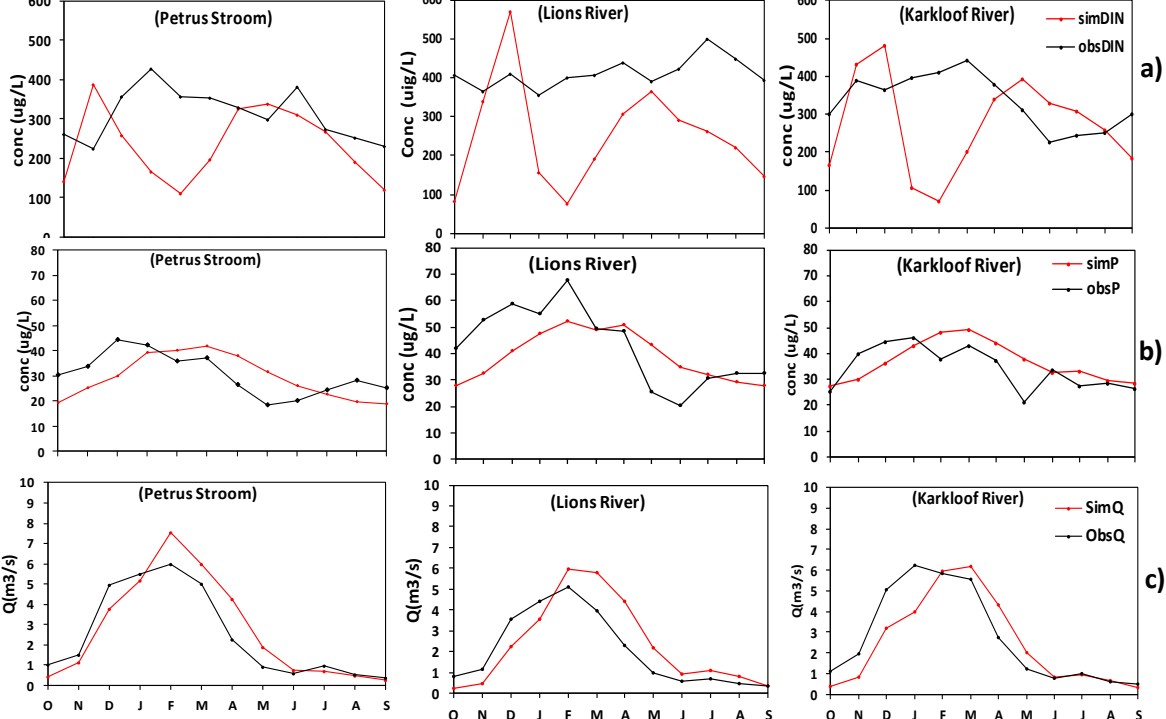

**Figure 8: Results of water modelling (seasonal distribution) of DIN in (a), TP (b) and flows (c), at Sub-catchments 7, 1510 and 2511, for the period (1989-1999). Sim stands for the simulated, Obs represents the observed, Q represents the streamflow, DIN represents the dissolved inorganic nitrogen and P represents the total phosphorus**





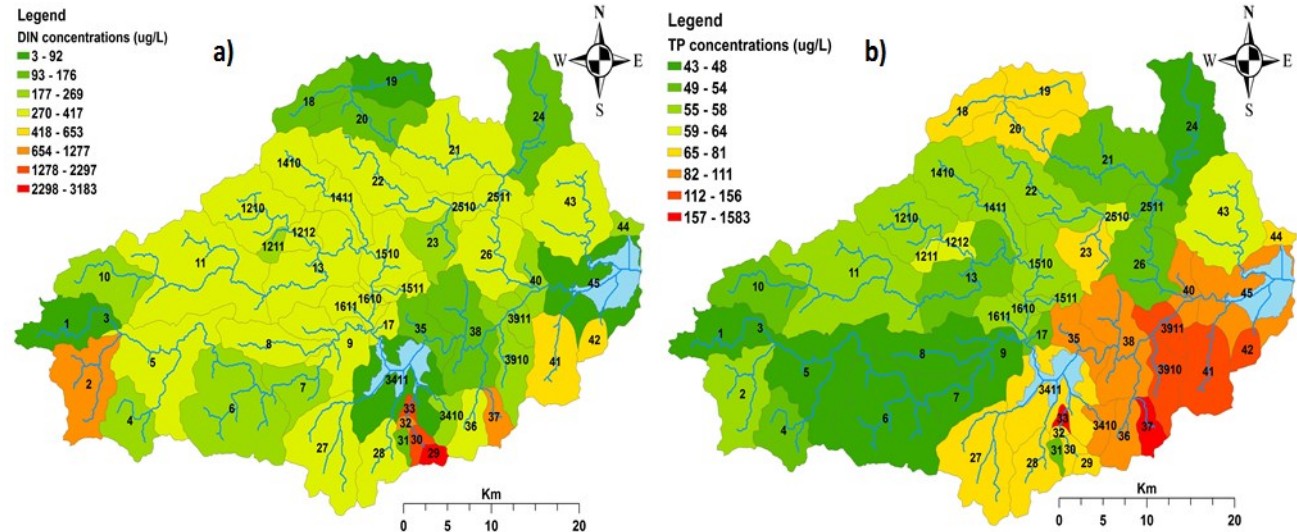

**Figure 9: Mean annual concentration of DIN (a) and TP (b) in the catchment for the period 1989-1999**






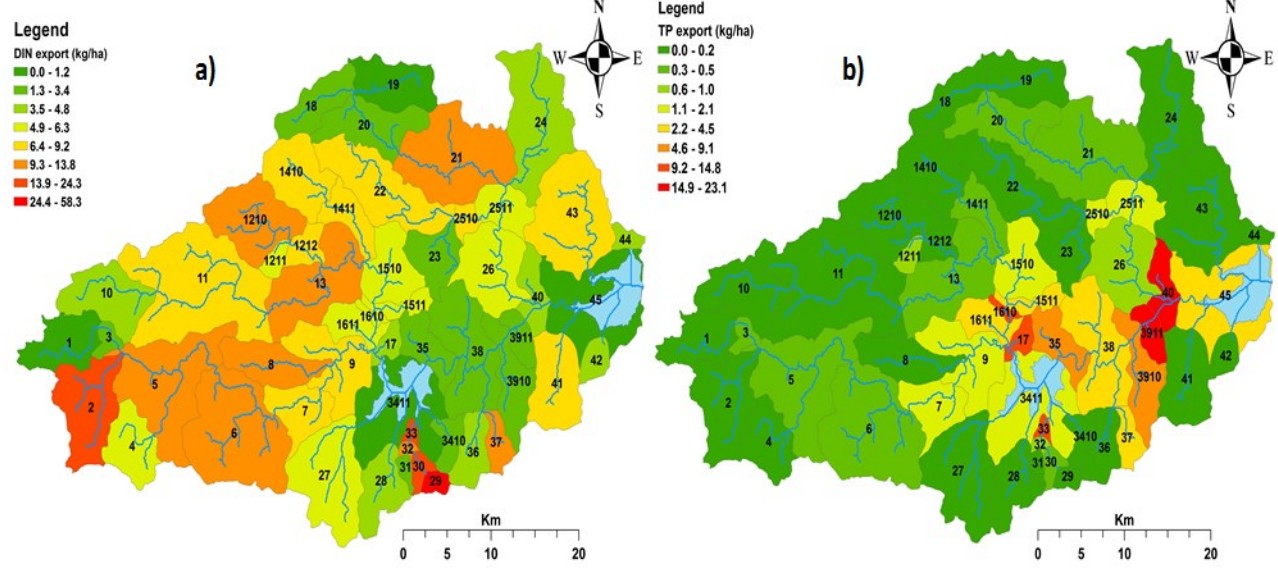

**Figure 10: Average area-weighted annual DIN (a) and TP (b) loads for each sub-catchment (1989-1999)**