# Peer review of "Assessment of the Hype Model for Simulation of Water and Nutrients in the Upper uMngeni River Catchment in South Africa"

_Hydrology and Earth System Sciences, 2017_

## Referee Comment (RC1) · Anonymous Referee #1 · 21 Aug 2017

The main objective of the work is to test the capability of the model HYPE to simulate streamflow, transport of dissolved inorganic nitrogen and total phosphorus which were observed in a river catchment in South Africa. The HYPE model requires an estimate of more than 100 parameters. The authors selected 25 of them (6 for streamflow and 19 for water quality) for standard manual model calibration. Because some of them are land use dependent, it is not clear to me if the number of calibrated parameters is 25 or if it is much more, related to the different land uses that exist in the catchment. After calibration, the authors provide a detailed and convincing discussion about the model results (comparison between modelled and simulated quantities).

[Figure]

The manuscript describes an interesting application of the model HYPE but misses the main objective of the work. Assessment of model for flow and transport requires not only a good match between computed and measured variables but also: - A detailed description of the main processes involved in the model. The presentation of the main features is too short. - A more relevant analysis of the model performance which is also depending on the number of calibrated parameters. The criteria used in this work (NSE, Pearson's correlation coefficient, percent bias) are not relevant for model assessment. - Detailed analyses of the calibrated parameter sets, including an estimate of the parameter uncertainty and parameters correlation. - A discussion on eventual over-parameterization (especially in case of significant correlation between some parameters). - A much more detailed discussion on the data set which mixes different time and space scales and different measurement errors.

Therefore, I consider that the paper should not be accepted for publication in HESS.

---

## Referee Comment (RC2) · Anonymous Referee #2 · 21 Aug 2017

This paper assesses the capability of the Hydrological Predictions for the Environment (HYPE) model in simulating streamflow, dissolved inorganic nitrogen (DIN) and total phosphorus (TP), in uMngeni Catchment in KwaZulu-Natal province, South Africa. The model was manually calibrated using stepwise approach and tested against observation and its performances were assessed based on the Nash-Sutcliffe efficiency (NSE), percent bias (PBIAS) and Pearson's correlation coefficients. Authors concluded that the Hype model was successful in simulating streamflow, DIN and TP in the upper uMngeni catchment.

The paper is a good application of the Hype model rather than an improvement in hy-

drological/nutrient processes understanding and modelling approach in general and in the region. There are many shortcomings in the manuscript, most importantly the lack of uncertainty analysis. I think that a sensitivity analysis and uncertainty assessment of the Hype model particularly to land use and soil parameterization could improve the quality of the paper. In addition, the paper misses a through discussion about the limitation of the Hype model for discharge and nutrient simulation in the uMngeni catchment. Authors just enumerated few of them in the conclusions but these statements are not directly supported by the findings of the paper.

Detailed comments - Land use and soil data were desegregated from coarser scale which could be not coherent with the hydrological model scale. The Hype model is quite sensitive to the land use and soil-type information. Does the scale and the resolution of these inputs affect the model performances?

- There is a wide panoply of techniques for automatic calibration of model parameter in literature which are faster and provide an insight on parameter sensitivity and uncertainty. So, what are the reasons behind using manual calibration in this work?

- P6, L, 181. Why a third thick soil layer was added during the calibration of the model?

- P7, L.213. The HYPE model has over one hundred parameters. How did you identified the most sensitive parameters for the calibration process?

---

## Author Comment (AC1) · 16 Oct 2017

**Introduction**

*The manuscript on "Assessment of the HYPE Model for simulation of water and nutrients in the upper uMngeni River Catchment in South Africa" was submitted to the journal as a case study. The paper aimed to assess the capability of the model in simulating stream flow and transport of nutrients (nitrogen and phosphorus) in a fast-developing catchment, typical of many in developing countries, with limitations in data available. This study was motivated by inclusion of in-stream processes of transformation of nutrients in waterbodies in the HYPE model which lack in the locally-developed model in South Africa, i.e. the ACRU-NPS and this is over-simplified in SWAT model. This study also aimed to assess the capability of the model to represent the processes driving water quality in the uMngeni Catchment.*

**Anonymous Referee #1**

**Comment 1.1**

The main objective of the work is to test the capability of the model HYPE to simulate streamflow, transport of dissolved inorganic nitrogen and total phosphorus which were observed in a river catchment in South Africa. The HYPE model requires an estimate of more than 100 parameters. The authors selected 25 of them (6 for streamflow and 19 for water quality) for standard manual model calibration. Because some of them are land use dependent, it is not clear to me if the number of calibrated parameters is 25 or if it is much more, related to the different land uses that exist in the catchment. After calibration, the authors provide a detailed and convincing discussion about the model results (comparison between modelled and simulated quantities).

**Response 1.1**

*The number of parameters used for the manual calibration is more than 25 and in total the authors have adjusted approximately 124 parameters during the calibration: for example, in total 42 parameters are general, the parameters which depend on land use are 10, those related to land use but affecting water quality were 23. The parameters which affect, soil, soil-water interactions and water quality in general are 12,15 and 22, respectively. However, the model is very sensitive to the 25 parameters presented in Appendix A.1 (the title has been changed to a list of most __sensitive__ and __all__ was deleted in the revised manuscript), but the calibration has considered more parameters. In the manuscript, the line 287 has been corrected to a list of __the most sensitive__ parameters (__all__ was replaced by __the most sensitive__)*

*These parameters were also highlighted in the other previous research studies that applied the HYPE Model, such as those of Jomaa et al. (2016), Jiang et al. (2014) and Yin et al. (2016).*

**Comment 1.2**

The manuscript describes an interesting application of the model HYPE but misses the main objective of the work. Assessment of model for flow and transport requires not only a good match between computed and measured variables but also: A detailed description of the main processes involved in the model. The presentation of the main features is too short. A more relevant analysis of the model performance which is also depending on the number of calibrated parameters.

**Response 1.2**

**These details on the main processes involved will be added in the revised manuscript**

*Above the ground, the model simulates snow, evapotranspiration, glaciers, rivers, lakes and routing. In the sub-catchments, air temperature is calculated using the average elevation, while precipitation is assumed to be uniform over each sub-catchment. In the HYPE model, potential evaporation is calculated from the air temperature and occurs when the air temperature is greater than a threshold temperature. It is assumed that evaporation from the soil decreases with depth and occurs in the two upper layers. The modeller can select one of the six methods of calculating potential evaporation in HYPE model, depending on the availability of input data.*

*Within the soil, water content is computed for each of a maximum of three layers. The location of the ground water table is calculated from the degree of soil saturation above field capacity in the different soil layers. Internal and outlet lakes and local and main rivers are defined in the model. Internal and outlet Lakes are taken as Soil and Land use Classes (SLC). The length of the local river in each sub-catchment may be given as an input to the model or is by default calculated as the square root of the sub-catchment land area. The delay in the river was determined as the length of the river (rivlen) and the water maximum velocity (rivvel). Soil runoff from the different soil layers depends on water content above field capacity and recession coefficients. Overland flow may occur if the top soil layer is over-saturated or if the infiltration capacity is exceeded. All local runoff waters from the SLC classes in a sub-catchment enter the local river from where it is routed directly to the main river or partially through an internal lake. The main river receives water from the local rivers in addition to flow from upstream sub-catchments. Water from the main river may pass an outlet lake before being discharged to the downstream sub-catchment (Lindstrom et al., 2010). Lakes are assumed to be completely mixed and for each lake a rating curve, area and depth are defined. Different reservoir regulation routines are also available. Water withdrawal for irrigation is considered as an important factor in water management and may be handled by the model.*

*Daily volumes and concentrations of nutrients from wastewater treatments plants, industries and inter-basin water transfer are considered as positive point sources, while water abstraction from lakes are defined as negative point sources. The water discharges from the rural households which are not connected to the municipal wastewater works are added to the internal river and to the deepest soil layers for each sub-catchment. Atmospheric deposition is recognised as an important source of nitrogen and phosphorus to land and lake classes. Wet deposition of nitrogen and phosphorus are added in the form of their respective concentrations of rainfall. Dry deposition of inorganic nitrogen is defined for each sub-catchment and are assumed to be land type dependent. Dry deposition of soluble phosphorus is given as a land use dependent parameter.*

*In the HYPE model the simulated processes that affect the nutrients in surface waters are denitrification, mineralisation, primary production and sedimentation. It is also assumed that there is*

*an exchange between particulate phosphorus in the water column and in sediments in the river. The width and the depth of a watercourse are important for the in-stream transformation of nutrients. These are calculated from a number of empirical equations. The bottom area of watercourse is calculated as the width times the length. The temperature of a water course is calculated by weighting the previous day water temperature and the air temperature.*

**Comment 1.3**

The criteria used in this work (NSE, Pearson's correlation coefficient, percent bias) are not relevant for model assessment.

*Response 1.3*

*Although it is not clear what the referee #1 meant by this, other criteria such as:*

- *the consideration of match between the simulated and the observed values,*
- *the graphical visualisation,*
- *comparison between the average daily simulated and measured flows and concentrations of nutrients*
- *the model capturing of peak and low flow events as drivers of pollution events,*
- *comparison of the seasonal distribution of average simulated and measured runoff and concentration of nutrients*
- *the calculation of water balance of the headwater sub-catchments*
- *maps of distribution of nutrient concentrations and loads and*
- *the literature on modelling of streamflow in the catchment and the knowledge of the author of the catchment were used.*

**Comment 1.4**

Detailed analyses of the calibrated parameter sets, including an estimate of the parameter uncertainty and parameters correlation. A discussion on eventual over-parameterization (especially in case of significant correlation between some parameters). A much more detailed discussion on the data set which mixes different time and space scales and different measurement errors

*Response 1.4*

*We recognised the risk of equifinality and we tried to overcome this by using a step-wise calibration approach by isolation of some processes using some stations, in sub-catchments without lakes, looking to the dominant land use and soil types. The choice of this method, regardless of being time consuming was to attain a better understanding and representation of hydrological and water quality processes and to avoid equifinality in model factors and parameters, rather than optimising Nash-Sutcliffe Efficiency (NSE). In addition, Moreover, due to a large number of parameters to calibrate, we thought that an automatic calibration was not much plausible than the manual calibration as indicated in other applications of the HYPE model (Lindstrom et al. 2010).*

*For hydrological part of the model:*

- *we started by calibrating the sub-catchments with the most common land use and soil, without lakes and isolation of the processes in lakes and rivers*
- *to achieve this, we first started with the general parameters that affect the water balance and flow discharge by looking to evaporation routine, i.e. the recession coefficient for surface runoff (srrcs) and the crop coefficient for PET model (Kc). Adjustments of the input data files*

*on precipitation and temperature provided in Pobs.txt and Tobs.txt, respectively and correction of the altitude.*

- *we continued with soil parameters which affect flow paths, dynamics of groundwater and discharge from headwaters i.e. water holding capacity, infiltration, percolation, recession and surface runoff (they also affect the concentration of nitrogen and phosphorus).*
- *the parameters which affect discharges in lakes and in river reaches (rivvel and damp)*
- *at the end, we added the isolated parameters in rivers and lakes*

*For water quality model, the calibration of the following parameters was carried out:*

- *The general parameters that control the denitrification processes in local and main rivers (denitwrl and denitwrm) and in lakes (denitwl). The fastN and fastP pools in soil and the factors affecting sedimentation and resuspension of particulate phosphorus (PP)in rivers (sedexp), as well as sedimentation rate of phosphorus and nitrogen in lakes (sedpp and sedon)*
- *The land use dependent parameters which guide the denitrification in all soil layers (denitrlu), the release of inorganic nitrogen from slowN via fastN. This parameter provides a steady release of IN. We also adjusted the land use dependent parameters that control the release of organic nitrogen (ON) from slowN (dissolhn, and minerfn) and the release of PP from slowP (dissolfn and minerfp).*
- *The soil dependent parameters adjusted are namely, freuc which controls the leaching concentrations of suspended phosphorus (SP), the resistance of soil to erosion due to overland flow (soilcoh) and the parameter affecting erosion caused by kinetic energy in rain (soilerod)*

*These processes allowed better understanding of the high retention of phosphorus which is characteristic of the soil in the uMngeni Catchment. They also helped to understand that phosphorus is mainly linked to the point sources of pollution, while dissolved inorganic nitrogen (DIN) is associated to both diffuse and point sources. This provided information on the possible sources of increased levels of nutrients of the river and impoundments and catchment-based knowledge on the transport and dynamics of nutrients in a catchment having a highly modified natural vegetation, with limitations in data available.*

---

## Author Comment (AC2) · 16 Oct 2017

*Introduction*

*The manuscript on "Assessment of the HYPE Model for simulation of water and nutrients in the upper uMngeni River Catchment in South Africa" was submitted to the journal as a case study. The paper aimed to assess the capability of the model in simulating stream flow and transport of nutrients (nitrogen and phosphorus) in a fast-developing catchment, typical of many in developing countries, with limitations in data available. This study was motivated by inclusion of in-stream processes of transformation of nutrients in waterbodies in the HYPE model which lack in the locally-developed model in South Africa, i.e. the ACRU-NPS and this is over-simplified in SWAT model. This study also aimed to assess the capability of the model to represent the processes driving water quality in the uMngeni Catchment.*

**Anonymous Referee #2**

**Comment 2.1**

This paper assesses the capability of the Hydrological Predictions for the Environment (HYPE) model in simulating streamflow, dissolved inorganic nitrogen (DIN) and total phosphorus (TP), in uMngeni Catchment in KwaZulu-Natal province, South Africa. The model was manually calibrated using stepwise approach and tested against observation and its performances were assessed based on the Nash-Sutcliffe efficiency (NSE), percent bias (PBIAS) and Pearson's correlation coefficients. Authors concluded that the HYPE model was successful in simulating streamflow, DIN and TP in the upper uMngeni catchment. The paper is a good application of the HYPE model rather than an improvement in hydrological/nutrient processes understanding and modelling approach in general and in the region. There are many shortcomings in the manuscript, most importantly the lack of uncertainty analysis. I think that a sensitivity analysis and uncertainty assessment of the HYPE model particularly to land use and soil parameterization could improve the quality of the paper. In addition, the paper misses a through discussion about the limitation of the HYPE model for discharge and nutrient simulation in the uMngeni catchment. Authors just enumerated few of them in the conclusions but these statements are not directly supported by the findings of the paper.

*Response 2.1*

*We disagree with this comment. Our approach does provide an improvement in understanding and modelling approach, especially in the region with limited data.*

*Other limitations of the model are:*

- *HYPE model uses static land use which is a challenge in the uMngeni Catchment, due its rapidly changing land use and modification of landscape*
- *simplification of the processes driving evapotranspiration in the model is a key challenge which affects the simulations of runoff in the catchment*
- *In the model the processes of inter-catchment transfer, water abstraction and release and atmospheric deposition of nutrients are static and over-simplified, while in reality they vary during the simulation period.*
- *Static used of daily volumes and concentrations of nutrients from the point sources of pollution, i.e. waste water treatment, industries*

**Comment 2.2**

Land use and soil data were desegregated from coarser scale which could be not coherent with the hydrological model scale. The Hype model is quite sensitive to the land use and soil-type information. Does the scale and the resolution of these inputs affect the model performances?

*Response 2.2*

*The desegregation was only made for soil data, while the land use data match the model scale. The authors recognised that the HYPE model depends on the description and parameterisation of land use and soil type information.*

**Comment 2.3**

There is a wide panoply of techniques for automatic calibration of model parameter in literature which are faster and provide an insight on parameter sensitivity and uncertainty. So, what are the reasons behind using manual calibration in this work?

*Response 2.3*

*The choice of the manual calibration of the model regardless of being time consuming was aimed to achieve a better understanding and representation of hydrological and water quality processes and to avoid equifinality in model factors and parameters. Moreover, due to a large number of parameters to calibrate, we thought that an automatic calibration was no more plausible than the manual calibration.*

*For hydrological part of the model:*

- *We started by calibrating the sub-catchments with the most common land use and soil, without lakes and isolation of the processes in lakes and rivers*
- *To achieve this, we first started with the general parameters that affect the water balance and flow discharge by looking to evaporation routine, i.e. the recession coefficient for surface runoff (srrcs) and the crop coefficient for PET model (Kc). Adjustments of the input data files on precipitation and temperature provided in Pobs.txt and Tobs.text, respectively and correction of the altitude.*
- *we continued with soil parameters which affect flow paths, dynamics of groundwater and discharge from headwaters i.e. water holding capacity, infiltration, percolation, recession and surface runoff (they also affect the concentration of nitrogen and phosphorus).*
- *the parameters which affect discharges in Lakes and in river reaches (rivvel and damp)*
- *At the end, we added the isolated parameters in rivers and lakes*

*For water quality model, the calibration of the following parameters was carried out:*

- *The general parameters that control the denitrification processes in local and main rivers (denitwrl and denitwrm) and in lakes (denitwl). The fastN and fastP pools in soil and the factors affecting sedimentation and resuspension of in rivers and in Lake (sedexp), as well as sedimentation rate of phosphorus and nitrogen in lakes (sedpp and sedon)*
- *The land use dependent parameters which guide the denitrification in all soil layers (denitrlu), the release of inorganic nitrogen from slow N via fastN. This parameter provides a steady release of IN. We also adjusted the land use dependent parameters that control the release of organic nitrogen (ON) from slow N (dissolhn, and minerfn) and the release of particulate phosphorus (PP) from slowP (dissolfn and minerfp).*

- The soil dependent parameters adjusted are namely, *freuc* which controls the leaching concentrations of suspended phosphorus (SP), the resistance of soil to erosion due to overland flow (soilcoh) and the parameter affecting erosion caused by kinetic energy in rain (soilerod)

*These processes allowed better understanding of the high retention of phosphorus which is characteristic of the soil in the uMngeni Catchment. They also helped to understand that phosphorus is mainly linked to the point sources of pollution, while dissolved inorganic nitrogen (DIN) is associated to both diffuse and point sources. This provided information on the possible sources of increased levels of nutrients of the river and impoundments and catchment-based knowledge on the transport and dynamics of nutrients in a catchment having a highly modified natural vegetation, with limitations in data available.*

**Comment 2.4**

- P6, L, 181. Why a third thick soil layer was added during the calibration of the model?

*Response 2.4*

*We started with the calibration of the two soil layers for which soil information in the catchment was available. Due to deep groundwater in the catchment, a third layer was added and an increase of the drain-depth in geoclass file data, in order to include the hydrological activities related to the groundwater runoff.*

**Comment 2.5**

-P7, L.213. The HYPE model has over one hundred parameters. How did you identify the most sensitive parameters for the calibration process?

*Response 2.5*

*The most sensitive parameters for the calibration process were identified in the literature review in the other applications of HYPE model (for example, Jiang et al. (2014), Jomaa et al. (2016) and Yin et al. (2016), by manual adjustment of the input parameters and measurements of the output values and expert knowledge from previous model applications.*